# Adherence to Nutritional Practice Guideline in Premature Infants: A Nationwide Survey in Taiwan

**DOI:** 10.3390/nu16183181

**Published:** 2024-09-20

**Authors:** Chi-Shiuan Ting, Po-Nien Tsao, Hung-Chieh Chou, Ting-An Yen, Hsin-Chung Huang, Chien-Yi Chen

**Affiliations:** 1Department of Pediatrics, National Taiwan University Hospital, National Taiwan University College of Medicine, Taipei 100, Taiwan; chishiuan0212@gmail.com (C.-S.T.); tsaopn@ntu.edu.tw (P.-N.T.); hcchou@ntu.edu.tw (H.-C.C.); anneofgb@gmail.com (T.-A.Y.); newborn.huang@gmail.com (H.-C.H.); 2Department of Pediatrics, China Medical University Children’s Hospital, China Medical University, Taichung 404, Taiwan

**Keywords:** neonatology, prematurity, parenteral nutrition, enteral nutrition, clinical practice

## Abstract

Objectives: This study aimed to assess the current neonatal nutritional practices in Taiwan and promote consensus on standardized protocols. Methods: An online questionnaire comprising 95 items on parenteral nutrition (PN) and enteral nutrition (EN) practices was distributed to neonatal care units across Taiwan via email between August and December 2022. The responses were compared with the recommendations from the European Society for Pediatric Gastroenterology Hepatology and Nutrition for preterm infant care. Results: Most of the 35 neonatal units, comprising 17 level III and 18 level II units, that participated in this study adhered to standard PN protocols; however, only 30% of units used protein-containing solutions as the initial fluid. Over half of the neonatal units provided calcium, phosphate, and magnesium at less than the recommended dosage. Trophic feeding commenced within 48 h in 88% of the units, with the mother’s milk used as the first choice. All the units preferred commencing advanced feeding at <25 mL/kg/day. Conclusions: Most nutrient protocols for preterm infants in neonatal units in Taiwan meet recent guidelines, but discrepancies such as lower mineral supplements in PN and a slower advancement of enteral feeding increase nutritional risk. These issues warrant further research.

## 1. Introduction

Inadequate nutritional support results in suboptimal growth and poor developmental outcomes in preterm neonates [1,2,3,4]. Therefore, it is important to implement clinical practice guidelines that sufficiently address the nutritional needs of preterm infants. The consensus continues to evolve with accumulating evidence. Early aggressive parenteral nutrition (PN) with protein and lipid supplementation has been recommended in recent years [3]. An adequate intake of protein and minerals via PN plays an important role in enabling optimal growth [5,6]. An early initiation of enteral feeding and increasing the intake of breast milk, including donor human milk, support gut maturation and prevent complications in preterm infants. The most recent guidelines on neonatal nutrition, encompassing both parenteral and enteral methods, were developed via the regular revision of nutritional guidelines over decades.

The European Society of Pediatric Gastroenterology, Hepatology, and Nutrition (ESPGHAN) revised the guidelines for PN of preterm infants in 2018, which are considered as the most credible reference [5,6,7,8,9,10,11,12,13]. They also issued a position paper on enteral nutrition (EN) for preterm infants in 2022 [14]. Clinical practice often varies by region and context despite these guidelines [15,16]. The Taiwan Society of Neonatology published “Recommendation on nutritional care of preterm infants in Taiwan” in 2015 for the detailed PN and EN policy, and it was revised in 2022, which follows the ESPGHAN recommendation [17]. However, clinical practices vary across institutions due to different reasons. 

This study aimed to evaluate the implementation of the policies regarding PN and EN for preterm infants with a very low birth weight (VLBW) in neonatal care units in Taiwan and compare these implementations with the ESPGHAN guideline. We aim to conduct a survey to assess the current neonatal nutritional practices across Taiwan and promote consensus on standardized protocols.

## 2. Materials and Methods

A nationwide survey was conducted in all hospitals registered in the Taiwan Neonatal Network and the Premature Baby Foundation of Taiwan, which comprise 52 neonatal care units involved in providing over 90% of care for prematurity. This study was approved by the Ethics Committee of the National Taiwan University Hospital. 

The nutritional survey was conducted using a closed-response online questionnaire designed in accordance with the ESPGHAN guidelines and clinical practices. Research Electronic Data Capture (RedCap) platform was used to conduct the survey. The questionnaire comprised 95 multiple-choice and open-ended questions subdivided into three domains. Domain I corresponded to the capacity of the neonatal units and comprised items pertaining to the number of beds in the NICU and observation room and the annual number of neonate admissions (including preterm and infants with VLBW). Domain II corresponded to PN and comprised items pertaining to the age at the time of initiation; initial dosage; and target intakes of carbohydrates, protein, lipids, minerals, and electrolytes. Domain III corresponded to the EN policy and comprised items pertaining to the priming schedule, rate of advanced feeding, and choice of feeding. In addition to these domains, items pertaining to the monitoring of metabolic bone disease in prematurity (MBDP) were included in this survey. The questionnaire was disseminated individually to the chief of each neonatal unit via electronic mail, and the responses based on the consensus and policy of each unit and the data were collected using REDCap 14.6.4 software. The data were excluded if a connection with the unit could not be achieved or no response was obtained by January 2023.

All the neonatal care units included in this study were classified as level II and level III units (classified according to the policy statement of the American Academy of Pediatrics) [18]. The data obtained from level II and level III neonatal care units were compared. The initiation and dosage of nutrients were analyzed and compared with the ESPGHAN guidelines on pediatric parenteral nutrition [5,6,7,8,9,10,11,12] and the World Health Organization (WHO) recommendations for the management of preterm or low-birth-weight infants [19] to determine compliance with local and global standards.

### Statistical Analysis

All statistical analyses were performed using Statistical Analysis Software 9.4 M7 and Microsoft Excel 21 statistics. Continuous variables, expressed as median, range, and interquartile range (IQR), were compared using the Wilcoxon rank sum test. Comparisons between the categorical variables of the groups were performed using Fisher’s exact test and Chi-square test owing to the small sample size. A *p*-value of <0.05 was considered statistically significant.

## 3. Results

A total of 35 of the 52 hospitals, including 17 of the 19 level III units (89.4%) and 18 of the 33 level II units (54.5%), responded to the survey before January 2023, yielding a response rate of 67.3%. Table 1 provides the annual number of preterm and VLBW infants admitted to these 35 hospitals. Approximately 20–25% of the infants born in Taiwan were delivered in these hospitals [20], accounting for 40% of preterm infants and 75% of VLBW infants. 

### 3.1. Parenteral Nutrition

Table 2 describes the results of the PN policy survey including fluid, carbohydrate, protein, lipid, and mineral intake.

#### 3.1.1. Fluids

Approximately 90% of the neonatal units initiated parenteral fluid intake at a dose of 70–90 mL/kg/day. The targeted fluid intake upon achieving stable growth was >140 mL/kg/day in nearly two-thirds of neonatal units. Most units used 10% dextrose solution without protein on the first day of life owing to the immediate unavailability of PN fluid in their units. This was particularly common in level II units, with 80% of the units using 10% dextrose solution.

#### 3.1.2. Carbohydrates

The dosage of glucose administered varied widely across different institutes. The target dose of glucose was <8 mg/kg/min in half of the level II units and 29% of level III units. More than 90% of the surveyed units reduced the glucose infusion rate (GIR) when the blood glucose levels exceeded 250 mg/dL, rather than 180 mg/dL (as recommended by the recent guideline) [12]. The lowest GIR used during hyperglycemia varied across the units, ranging from 1.0 to 4.0 mg/kg/min. The glucose infusion was suspended in one unit if persistent hyperglycemia was observed. Most neonatologists in Taiwan tended to avoid the use of insulin in preterm infants.

#### 3.1.3. Proteins

The current guideline recommends administering protein supplements in PN on the first postnatal day [3]. However, protein supplementation was commenced within the first postnatal day in only half of the surveyed units. The proportion of level III units (73%) was significantly higher than that of the level II units (50%). The target protein intake in infants with VLBW varied across units, with most units administering doses higher than those recommended by the current guideline. Approximately 40% of the units administered protein at a dose of >3.5 g/kg/day in preterm infants weighing 1000–1500 g at birth. Approximately 80% of the units reported a target protein intake of >3.5 g/kg/day for preterm infants with an extremely low birth weight (ELBW); no significant differences were observed between levels of care.

#### 3.1.4. Lipids

Nearly 60% of the neonatal units initiated the administration of intravenous lipid emulsions on the second day of life. However, two level II units did not administer lipids to preterm neonates. The target dose for lipids, ranging from 2.0 to 3.9 g/kg/day, varied across units. Approximately half of the units aimed for a target lipid intake of 3.0–3.4 g/kg/day for infants with VLBW. Over 80% of the units administered SMOFlipid as lipid emulsions. However, two units still used a pure soybean oil-based lipid emulsion (INTRALIPID^®^).

#### 3.1.5. Mineral Intake

The implementation of the calcium (Ca) supplementation policy across the units was consistent, with two-thirds of the neonatal units administering Ca at a dose of 40–79 mg/kg/day. However, compared with level II units, level III units tended to aim for a significantly higher target phosphate (P) intake, ranging from 40 to 59 mg/kg/day (*p* < 0.05). Nearly 60% of the neonatal units reported a magnesium (Mg) intake of <5 mg/kg/day in both groups. Recent guidelines from the ESPGHAN [6] recommend administering 64–140 mg/kg/day of Ca and 50–108 mg/kg/day of P in neonatal PN regimens to facilitate optimal growth and bone mineralization in preterm infants. Most units did not achieve the optimal target. Approximately 80% of the surveyed units used organic calcium as a supplement, with the prevalence of its use being higher in level III units. However, organic phosphate was used in only 40% of the neonatal care units, comprising approximately 60% of the level III units and 22% of the level II units. The proportion of level III units was significantly higher than that of the level II units (*p* < 0.05).

#### 3.1.6. Monitoring

Some complications may arise during the use of parenteral nutrition, such as hypoglycemia or hyperglycemia, electrolyte imbalance, hyperlipidemia, and PN-associated liver disease. Therefore, the regular monitoring of associated parameters is suggested. Table 3 describes the monitoring schedule in preterm neonates using PN. Approximately 90% of the neonatal units regularly monitored nutrition-related biomarkers including the blood glucose, liver enzymes, electrolyte balance, and triglyceride levels. Electrolyte balance was the most frequently monitored biomarker, with 80% of the units monitoring it twice weekly or weekly. Approximately 70% of the units monitored the triglyceride levels weekly or biweekly, whereas nearly 70% of the departments monitored the liver enzyme levels on a weekly basis. Over 70% of the units did not monitor the coagulopathy factor levels regularly unless there were signs of bleeding.

### 3.2. Enteral Nutrition

Table 4 describes the survey results of enteral nutrition policy for infants with VLBW, including the choice of initial feeding, advancement of enteral feeding, and hospitalization for enteral nutrition for preterm infants.

#### 3.2.1. Choice of Initial Feeding

Buccal colostrum was administered to premature infants initially in almost 60% of the neonatal units, with level III units (70%) constituting the majority. Approximately 90% of units initiated minimal enteral feeding within two days of birth, with mother’s own milk (MOM) being the preferred choice in all units. However, donor human milk (DHM) was used as the second-line trophic feeding option in 70% of cases in level III units, compared to 50% in level II units. Additionally, half of the level II units chose 20 kcal/fl oz premature formula as their second-line option. 

#### 3.2.2. In-Hospital Enteral Nutrition

Approximately 70% of the surveyed units defined full feeding as ranging from 150 to 159 mL/kg/d for growing preterm infants. However, approximately 50% of level III units administered <150 mL/kg/d to infants who were completely dependent on EN. The advancement of enteral feed varied widely across the units, with 70% of the units selecting a slow advancement of 10–19 mL/kg/d. The 2022 WHO recommendation [19] states that the fast advancement (>30 mL/kg/day) of enteral feeding may reduce the time to full enteral feeding and the length of hospital stay. The majority of neonatal units tend to fortify the breast milk when the intake reaches 100 mL/kg/d. TPN was discontinued when the oral intake reached 120 mL/kg/d. Most neonatal units defined the standard of adequate weight gain as 15–24 g/kg/day after stable nutritional supplementation. More than 90% of the units implemented regular vitamin D supplementation for preterm infants. Thirty-two of the thirty-five surveyed units routinely monitored gastric residuals before every meal. However, 23 of these units continued feeding despite the presence of gastric residual volume. Over half of the units discontinued feeding if abdominal distension was suspected or an umbilical artery catheter was placed. Only 57% of the units, comprising 70% of level III units and 44% of level II units, implemented standardized feeding protocols.

#### 3.2.3. Survey of Metabolic Bone Disease of Prematurity

As shown in Table 5, 28 units, including 16 level III units (94.1%) and 12 level II units (66.67%), routinely screened for MBDP before discharge. All units commenced MBDP screening not later than 5 weeks of birth, and 57% of the units performed the first biochemical test between 4 and 5 weeks of birth. The majority of the units utilized the serum calcium, phosphate, and alkaline phosphatase (ALP) levels as screening biomarkers. However, <40% of the units utilized additional biomarkers, such as 25-OH vitamin D, intact parathyroid hormone (iPTH), and urine biomarkers, for further screening. Four units performed bone mass measurements on premature infants.

## 4. Discussion

A comprehensive survey of the prevailing objectives pertaining to PN and EN for infants with VLBW in Taiwan was conducted in this study. Compared with the most recent guidelines for neonatal nutrition of infants with VLBW issued by the ESPGHAN [5,6,7,8,9,10,11,12], most objectives met the guidelines. However, certain items continued to differ between the international guidelines and the policies in Taiwan; these differences may be attributed to regional variations. In Taiwan, most neonatologists remain concerned about the association of feeding and the occurrence of NEC and tend to be more conservative in their feeding strategies. Limited medical resources is also one of the key factors affecting the compliance with recommendations. Although the National Health Insurance covers most of the high medical costs for preterm infants in NICU, the system imposes certain limitations on clinical care options. For example, some neonatal units cannot provide protein on the first day or higher calcium and phosphate intake by PN due to inadequate resources. 

Over half of the surveyed units administered parenteral amino acid to preterm infants before the first day of life. Current guidelines recommend commencing the administration of amino acid intake in neonates from the first day of life to enhance protein synthesis while avoiding a decrease in proteolysis [21] as this approach can improve short-term growth [22,23]. However, the estimation of the required amino acid intake remains controversial. Almost 40% of the units surveyed in the current study administered >3.5 g/kg/day of parenteral amino acid to infants weighing 1000–1500 g. Over 80% of the units administered >3.5 g/kg/day of parenteral amino acid to infants with VLBW. The 2005 ESPGHAN guidelines recommend maintaining a minimum intake of 1.5 g/kg/day to prevent negative nitrogen balance, with a maximum of 4 g/kg/day [24]. Nevertheless, the evidence to support the contention that increasing the amino acid intake beyond 2.5 g/kg/day yields more favorable outcomes is limited. The effect of higher parenteral amino acid intake on long-term growth or neurodevelopment did not differ significantly [22,25,26]. Furthermore, high amino acid intake might elevate the urea concentration and sepsis rate. The 2018 ESPGHAN guidelines recommend maintaining the parenteral amino acid intake between 2.5 and 3.5 g/kg/day in preterm infants [5]. According to the recent guideline, the protein intake in this study was higher than what is currently recommended. This may be attributed to adhering to the previous 2005 ESPGHAN guideline. Nevertheless, the evidence for the influence of higher protein administration in preterm infants remains lacking.

The current study revealed that <50% of the units supplemented glucose infusion up to 8–10 mg/kg/min. In contrast, approximately 50% of the level II units aimed to maintain the infusion rate at <8 mg/kg/min. The optimal carbohydrate intake is determined based on the energy requirements, blood glucose levels, and growth. The glucose intake should be increased stepwise over 2–3 days, usually up to 10 mg/kg/min to facilitate growth thereafter. The ESPGHAN recommends that the parenteral carbohydrate intake should preferably not be >12 mg/kg/min or <4 mg/kg/min in preterm infants [12]. The blood glucose level is an important factor affecting the dose of glucose to be administered on the first postnatal day. Over 90% of the neonatal units adjusted the glucose infusion rate in response to blood glucose levels exceeding 250 mg/dL, with the minimum infusion rates varying across units. Hyperglycemia, defined as a blood glucose level of >180 mg/dL in preterm infants [27], is associated with increased morbidity [28,29,30,31]. However, the strict control of the blood glucose levels in critically ill children did not decrease mortality in previous studies [32,33]. Insulin therapy is effective in treating or preventing hyperglycemia in preterm infants, but it can lead to an increase in the incidence of hypoglycemia [34]. A general consensus on the management of blood sugar levels in premature infants has not been established in Taiwan, and the policy regarding the use of insulin in neonatal units remains unclear, warranting further research. 

The target intake of Ca and Mg was similar across all neonatal care units in Taiwan in the current study. The target intake of phosphate was significantly higher in level III units. Ca, P, and Mg, which constitute 98%, 80%, and 65% of the body content, respectively, are major components of the skeleton. Optimal PN should provide a slight surplus of Ca and P to ensure optimal tissue and bone mineral accretion [6]. The fetal total body analysis suggested that the theoretical optimal molar Ca and P ratio in PN for achieving fetal body composition in stable growing infants is 1.3 [6]. Recent guidelines from the ESPGHAN recommend administering 64–140 mg/kg/day of Ca and 50–108 mg/kg/day of P as a part of neonatal PN regimens to facilitate optimal growth and bone mineralization in preterm infants [6]. Most neonatal units did not achieve the mineral intake levels recommended by the ESPGHAN recommended mineral intake. This may be attributed to the lower doses recommended by the previous guideline in Taiwan (Ca: 60–80 mg/kg/day, P: 45–60 mg/kg/day) [17,35]. This difference may be attributed to the previous absence of a published guideline for mineral intake in PN and the compatibility issues of inorganic salts in PN. Inadequate mineral intake is associated with MBDP, which may cause osteopenia and severe bone disease with fracture [36,37,38]. 

The use of organic phosphate (NaGP) in PN solutions can improve compatibility and prevent precipitation [39,40,41]. Only 40% of the surveyed neonatal units administered organic phosphate, possibly owing to the Taiwan Food and Drug Administration approving the use of NaGP in 2022. Although NaGP in PN solutions has been used to increase Ca and P intake in preterm infants receiving PN support, it does not facilitate adequate Ca and P intake in infants with ELBW [41]. Hsu et al. suggested that this result may be attributed to the high sodium content in NaGP, which limits the total P intake. An inappropriately high sodium intake may result in fluid retention and increase the risk of complications, such as hemodynamically significant patent ductus arteriosus, bronchopulmonary dysplasia (BPD), retinopathy of prematurity, and intraventricular hemorrhage, in preterm infants [41]. 

Most of the surveyed neonatal units implemented a similar strategy of preterm EN including time to start feeding, choice of trophic feeding, and rate of advance feeding. EN aims to support optimal growth and development of preterm infants while preventing the incidence of complications such as necrotizing enterocolitis (NEC), catheter-related complications, infections, and sepsis [42,43]. The ESPGHAN advises commencing enteral feeding at the earliest, preferably within the first few hours of life, and gradually increasing feedings to achieve full feeds by 2–3 weeks of age [14]. Human milk is the preferred feeding option for preterm infants, followed by donor milk or 24 kcal/fl oz premature formula if human milk is unavailable. Fortifying breast milk or premature formula with additional nutrients aids in meeting the elevated nutritional requirements of preterm infants. Approximately 90% of the surveyed units initiated minimal enteral feeding within two days of birth, and MOM was the first choice in all units.

Most of the surveyed units adopted similar feeding policies; however, certain differences were observed between the studies [14,19]. Several trials have indicated that fast advancement (>30 mL/kg/day) of enteral feeding may reduce the time to full enteral feeding and length of hospital stay without increasing the incidence of NEC [44,45]. However, all surveyed units preferred implementing a slow advancement policy for preterm infants. Over 50% of the units included in the current study preferred a rate of 10–19 mL/kg/day. This result could be attributed to the neonatologists in Taiwan being cautious about the risk of NEC and setting their own guidelines based on the findings of previous study [45]. The prevalence of NEC may be lower in Taiwan compared with that in other parts of the world owing to the implementation of the slow advancement policy. A global meta-analysis conducted in 2020 [46] showed that seven out of one hundred infants with VLBW (1.5–17%) were diagnosed with NEC and that the incidence rate of NEC from 2016 to 2021 was around 4.5% in Taiwan, which is slightly lower than the global incidence rate. Nevertheless, the correlation between fast advancement and the incidence of NEC remains unclear, warranting further studies. Rapid advancement in enteral feeding and the optimization of standard nutrition protocols are associated with potential benefits, such as increased body weight gain, decreased time to achieve full feeding, and reduced requirement for the prolonged use of PN [45,47]. These may help reduce the length of hospital stay and the incidence of PN-related complications, potentially benefiting neonatal neurodevelopment.

Twenty-eight of the thirty-five surveyed neonatal units screened for MBDP regularly; the screening commenced within 5 weeks. MBDP, a common complication observed in preterm infants recently, is characterized by suboptimal bone matrix mineralization and biochemical alterations of phospho-calcium metabolism. A lower birth weight, prolonged use of PN, and the incidence of NEC and BDP were identified as risk factors for MBDP [48]. The clinical signs of MBDP, which appear between 5 and 11 weeks of life, are characterized by an increased work of breathing (owing to chest wall instability caused by softening or fractures of ribs), an enlargement of the cranial sutures, frontal bossing, rickets, fractures, and postnatal growth failure [49,50]. The assessment of serum biochemical markers, such as Ca, P, ALP, iPTH, and vitamin D, can facilitate the early detection of mineral deficiency caused by MBDP [51,52,53]. The prevention strategy comprises improving nutrition, specifically the intake of Ca, P, and vitamin D, and limiting the chronic use of diuretics and methylxanthines that reduce mineral stores and glucocorticoids that enhance bone resorption. A biweekly monitoring of the above mentioned biochemical markers should be performed in infants at high risk of developing MBDP [48,54]. The long-term consequences of MBDP are difficult to study, partly owing to the difficulties in defining a diagnosis of MBDP and the lack of evidence regarding the correlation between MBDP and growth restriction or the incidence of fracture in childhood and adulthood [54]. Over 80% of the surveyed 28 units monitored the Ca, P, and ALP levels within a month; however, the vitamin D and iPTH levels were assessed less frequently. Previous studies have shown that the lack of mineral administration in PN, a slow advancement of feeding policy, and prolonged use of PN may increase the risk of developing MBDP. Diagnostic criteria for MBDP remains to be established; however, routine monitoring remains crucial among premature infants in Taiwan owing to the multitude of high-risk factors associated with our nutrition policy. However, the influence of MBDP and the complication warrant more evidence.

In summary, we have highlighted several key differences between international guidelines and current clinical practice in Taiwan. First, only 30% of units provide protein-containing solutions as the initial fluid. This is a systemic issue that requires more resources to improve. Secondly, over half of the neonatal units provided calcium, phosphate, and magnesium at less than the recommended dosage. The inadequate mineral intake in PN may contribute to premature osteopenia and increased fracture risk. Promoting the use of organic calcium and phosphate salts in PN and revising the existing guideline in Taiwan may improve the compliance. Finally, the practice of slow advancement of feeding is common in Taiwan, possibly due to the concerns of the risk of NEC. This feeding policy may lead to prolonged PN usage and extended hospital stays [45,47]. In our opinion, rather than applying a uniform slow feeding approach for all infants, feeding advancement should be customized based on clinical markers, such as gut function and tolerance. 

Certain limitations exist to this study. First, the questionnaires were sent to the chiefs of the units, not the individual neonatologists. The variation between the nutritional objectives followed by each physician may have been higher than that observed between the nutritional protocols. However, the response from each unit represents the nutritional policy from the consensus formed among their medical staff. The result reflects the clinical practice of the majority of healthcare professionals. If the survey is collected on a per-person basis, the clinical practice of neonatal units with fewer doctors may be overlooked. Another limitation is that the charts of the infants were not reviewed to document actual feed. A comparison between the perceived and actual practices may reveal sizable variations [55]. There have been significant advancements in the clinical care of nutrition for preterm infants recently, but some practices have not yet reached a consensus or remain controversial. This questionnaire was developed in accordance with the ESPGHAN guidelines and focuses solely on the general principles of PN and EN care. Some issues of care that have not yet reached a consensus or remain controversial are not included in this survey. Some topics that have not yet reached a consensus in guideline were not included in this survey, such as the use of probiotics or in NICU. 

## 5. Conclusions

In conclusion, the findings of the current study describe the PN and EN protocols for preterm infants in neonatal units in Taiwan. Most of these policies met the recent guidelines; however, certain discrepancies were observed between these practices and the recent policy. First, the target mineral intake of PN was lower than the national recommendation, especially in the level II units. Second, the advancement of enteral feeding policy tended to be slower (10–19 mL/kg/day) than what is recommended by the guidelines. These differences are risk factors for MBDP that warrant further research.

## Figures and Tables

**Table 1 nutrients-16-03181-t001:** Demographic characteristics of respond institutions.

	Level III Units	Level II Units	Total
Number of units included in the study			
Units responded to questionnaire	17	18	35
Bed number of neonatal intensive care unit	15 (8–50)	6 (0–16)	435
Bed number of observation room	25 (12–54)	12 (5–22)	746
Number of neonatologists	5 (1–15)	2 (1–8)	133
Number of neonates per year			
Annual numbers of neonates in Taiwan			163,484
Admissions number of neonates	1300 (550–3168)	675 (367–1400)	38,694 (23.66%)
Annual numbers of prematurity in Taiwan			16,990
Admissions number of prematurity	230 (55–800)	72.5 (20–200)	6697 (39.4%)
Annual numbers of VLBW in Taiwan			1644
Admissions number of VLBW	45 (4–222)	7.5 (1–50)	1248 (75.9%)

This table is represented as the median number (Range) or number (percentage).

**Table 2 nutrients-16-03181-t002:** Parenteral nutrition policy for preterm infants.

	Total		Level III Units		Level II Units		*p*
Initial fluid intake							
70–90 mL/kg/day	32	(91%)	15	(88%)	17	(94%)	0.735
>90 mL/kg/day	1	(3%)	1	(6%)	0	(0%)	
Other	2	(6%)	1	(6%)	1	(6%)	
Target fluid intake							
<140 mL/kg/day	12	(34%)	6	(35%)	6	(33%)	1.00
>140 mL/kg/day	23	(66%)	11	(35%)	12	(67%)	
Initial fluid choice							
10% glucose	25	(71%)	11	(65%)	14	(78%)	0.47
Parenteral nutrition	10	(29%)	6	(35%)	4	(22%)	
Initial glucose intake							
<4 mg/kg/min	1	(3%)	1	(6%)	0	(0%)	0.485
4–8 mg/kg/min	34	(97%)	16	(94%)	18	(100%)	
Target glucose intake							
<8 mg/kg/min	14	(40%)	5	(29%)	9	(50%)	0.458
8–10 mg/kg/min	15	(43%)	8	(47%)	7	(39%)	
>10 mg/kg/min	4	(11%)	2	(12%)	2	(11%)	
Clinically	2	(6%)	2	(12%)	0	(0%)	
Age to initiated protein intake							
<24 h	22	(63%)	13	(76%)	9	(50%)	0.376
0–48 h	10	(28%)	3	(18%)	7	(39%)	
24–48 h	3	(9%)	1	(6%)	2	(11%)	
Target protein intake for birth weight ranging from 1001 to 1500 g							
<2.5 g/kg/day	1	(3%)	1	(6%)	0	(0%)	0.733
2.5–3.5 g/kg/day	20	(57%)	10	(59%)	10	(56%)	
>3.5 g/kg/day	14	(40%)	6	(35%)	8	(44%)	
Target protein intake for birth weight of <1000 g							
<2.5 g/kg/day	0	(0%)	0	(0%)	0	(0%)	1.00
2.5–3.5 g/kg/day	7	(20%)	3	(18%)	4	(24%)	
>3.5 g/kg/day	27	(80%)	14	(82%)	13	(76%)	
Age of initial lipid intake							
<24 h	4	(11%)	2	(12%)	2	(11%)	0.927
24–48 h	22	(63%)	12	(70%)	10	(56%)	
48–72 h	5	(14%)	2	(12%)	3	(17%)	
Clinically	2	(6%)	1	(6%)	1	(6%)	
Not given	2	(6%)	0	(0%)	2	(11%)	
Target lipid intake for birth weight ranging from 1001–1500 g							
2.0–2.4 g/kg/day	4	(12%)	3	(18%)	1	(6%)	0.804
2.5–2.9 g/kg/day	11	(33%)	6	(35%)	5	(31%)	
3.0–3.4 g/kg/day	16	(48%)	7	(41%)	9	(56%)	
3.5–3.9 g/kg/day	2	(7%)	1	(6%)	1	(6%)	
Target lipid intake for birth weight of <1000 g							
2.0–2.4 g/kg/day	3	(9%)	2	(12%)	1	(7%)	1.00
2.5–2.9 g/kg/day	9	(28%)	5	(29%)	4	(27%)	
3.0–3.4 g/kg/day	17	(53%)	9	(53%)	8	(53%)	
3.5–3.9 g/kg/day	3	(9%)	1	(6%)	2	(13%)	
Target Calcium intake							
20–39 mg/kg/day	2	(9%)	0	(0%)	2	(11%)	0.399
40–59 mg/kg/day	11	(28%)	4	(24%)	7	(39%)	
60–79 mg/kg/day	15	(53%)	9	(53%)	6	(33%)	
>80 mg/kg/day	3	(9%)	1	(6%)	2	(11%)	
Clinically	4	(11%)	3	(17%)	1	(6%)	
Organic Calcium used	27	(77%)	15	(88%)	12	(67%)	0.22
Target Phosphate intake							
<19 mg/kg/day	2	(6%)	0	(0%)	2	(11%)	<0.05
20–39 mg/kg/day	10	(29%)	3	(18%)	7	(39%)	
40–59 mg/kg/day	17	(49%)	11	(64%)	6	(33%)	
Clinically	3	(8%)	3	(18%)	0	(0%)	
Not given	3	(8%)	0	(0%)	3	(17%)	
Organic Phosphate used	14	(40%)	10	(59%)	4	(22%)	<0.05
Target Magnesium intake							
<5 mg/kg/day	21	(60%)	11	(65%)	10	(56%)	0.084
<5 mg/kg/day	8	(23%)	4	(24%)	4	(22%)	
Clinically	6	(17%)	2	(12%)	4	(22%)	

**Table 3 nutrients-16-03181-t003:** The monitor policy during use of parenteral nutrition in preterm infants.

	Once Daily		Twice a Week		Weekly		Biweekly		Monthly		Clinically		Total
Blood sugar	5	(16%)	8	(25%)	9	(28%)	0	(0%)	1	(3%)	9	(28%)	32
Electrolyte	1	(3%)	12	(38%)	15	(47%)	2	(6%)	1	(3%)	1	(3%)	32
Triglyceride	0	(0%)	0	(0%)	14	(44%)	9	(28%)	5	(16%)	4	(12%)	32
Liver function	0	(0%)	1	(3%)	23	(72%)	5	(16%)	3	(9%)	0	(0%)	32
Coagulation	0	(0%)	0	(0%)	2	(6%)	3	(10%)	3	(10%)	23	(74%)	31

**Table 4 nutrients-16-03181-t004:** Enteral nutrition policy for preterm infants.

	Total		Level III Units		Level II Units		*p*
Start of trophic feeding							
<48 h	31	(89%)	16	(94%)	15	(83%)	0.602
48–96 h	4	(11%)	1	(6%)	3	(17%)	
Colostrum as mouth care							
Yes	22	(63%)	12	(71%)	10	(56%)	0.488
Choice of trophic feeding							
MOM only	1	(3%)	1	(6%)	0	(0%)	0.16
MOM > DM	21	(60%)	12	(70%)	9	(50%)	
MOM > PF	13	(37%)	4	(24%)	9	(50%)	
Definition of full feeding							
<140 mL/kg/day	8	(23%)	4	(24%)	4	(22%)	0.102
140–149 mL/kg/day	4	(11%)	4	(24%)	0	(0%)	
150–159 mL/kg/day	23	(66%)	9	(52%)	14	(78%)	
Rate of advance feeding							
1–9 mL/kg/day	5	(14%)	3	(18%)	2	(11%)	0.77
10–19 mL/kg/day	24	(69%)	12	(70%)	12	(67%)	
20–25 mL/kg/day	6	(17%)	2	(12%)	4	(22%)	
Time to add HMF							
<60 mL/kg/day	2	(6%)	0	(0%)	2	(11%)	0.396
60–80 mL/kg/day	4	(11%)	1	(6%)	3	(16%)	
80–100 mL/kg/day	15	(43%)	9	(53%)	6	(34%)	
100–120 mL/kg/day	13	(37%)	6	(35%)	7	(39%)	
Other	1	(3%)	1	(6%)	0	(0%)	
Time to stop PN							
<100 mL/kg/day	2	(5%)	0	(0%)	2	(11%)	0.632
100–120 mL/kg/day	17	(49%)	9	(53%)	8	(44%)	
120–140 mL/kg/day	14	(40%)	7	(41%)	7	(39%)	
>140 mL/kg/day	1	(3%)	0	(0%)	1	(6%)	
Other	1	(3%)	1	(6%)	0	(0%)	
Expected weight gain							
<14 g/kg/day	2	(6%)	1	(6%)	1	(6%)	0.470
15–24 g/kg/day	31	(88%)	14	(82%)	17	(94%)	
>25 g/kg/day	2	(6%)	2	(12%)	0	(0%)	
Regular vitamin D supply							
Yes	31	(89%)	16	(94%)	15	(83%)	0.602
Establish feeding protocol							
Yes	20	(57%)	12	(70%)	8	(44%)	0.175

**Table 5 nutrients-16-03181-t005:** The monitor policy metabolic bone disease of prematurity.

Weekly		Biweekly		Monthly		Bimonthly		Not recorded		Clinically		Total
Alkaline Phosphatase												
0	(0%)	16	(57%)	11	(39%)	1	(3.5%)	0	(0%)	0	(0%)	28
Serum calcium/phosphate												
3	(11%)	16	(57%)	8	(29%)	1	(3.5%)	0	(0%)	0	(0%)	28
25-OH Vitamin D												
0	(0%)	1	(4%)	5	(18.5%)	1	(4%)	15	(56%)	5	(18.5%)	27
Intact parathyroid hormone												
0	(0%)	1	(3.5%)	5	(18%)	0	(0%)	17	(60%)	5	(18%)	28
Urine calcium/phosphate												
0	(0%)	1	(3.5%)	4	(14%)	0	(0%)	19	(68%)	4	(14%)	28
Radiography												
0	(0%)	6	(21%)	10	(36%)	3	(11%)	5	(18%)	4	(14%)	28
Bone Mass Measurement												
0	(0%)	1	(3.5%)	1	(3.5%)	0	(0%)	24	(86%)	2	(7%)	28

## Data Availability

The original contributions presented in the study are included in the article, further inquiries can be directed to the corresponding author.

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
