# Peer review of "Adherence to Nutritional Practice Guideline in Premature Infants: A Nationwide Survey in Taiwan"

_nutrients, 2024, doi:10.3390/nu16183181_

Round 1

Reviewer 1 Report

Comments and Suggestions for Authors

Clearly nutrition and nutritional intervention are of paramount importance in premature infants and so data that examines compliance with international recommendations should be welcomed.

The introduction provides a good review of the role of nutrition in the care of premature infants and also provides background information relating to one set of international recommendations.

The methods section is straightforward in describing a relatively simple approach to assessing compliance to recommendations in some 35 neonatal units.

The results section provides a great deal of information covering a huge range of nutritional parameters.

There are a few changes or additions to manuscript that I think would be beneficial.

Discussion Page 8 line 207.  I would like the authors to expand and explain what the “regional differences” are that they believe are having an influence on compliance with recommendations. 

I also think that in light of the vast amount of data presented it might be useful in the discussion for the authors to add a paragraph that states which of the many findings the authors feel are the, say, 5 most important and why. 

There could also be some suggestions from the authors in the discussion how the variation in practice and some of the practices that lead to non compliance could be addressed nationally within Taiwan.

Finally there are a number of spelling mistakes that should be corrected throughout the manuscript.

Comments on the Quality of English Language

Some spelling errors to be corrected.

Author Response

Comment 1 : Discussion Page 8 line 207. I would like the authors to expand and explain what the “regional differences” are that they believe are having an influence on compliance with recommendations.

Response 1: Thank you for your thoughtful comments. We have carefully considered the suggestion and expand the discussion about the regional difference on page 9 line 211-219, as followed. 

“However, certain items continued to differ between the international guidelines and the policies in Taiwan; these differences may be attributed to regional variations. In Taiwan, most neonatologists remain concerned about the association of feeding and occurrence of NEC, and tend to be more conservative in their feeding strategies. Limited medical resource is also one of the key factors affected the compliance with recommendations. Although the National Health Insurance covers most of the high medical costs for preterm infants in NICU, the system imposes certain limitations on clinical care options. For example, some neonatal units cannot provide protein on the first day or higher calcium and phosphate intake by PN due to inadequate resources. ”

Comment 2: I also think that in light of the vast amount of data presented it might be useful in the discussion for the authors to add a paragraph that states which of the many findings the authors feel are the, say, 5 most important and why.

Response 2: We sincerely appreciate your valuable comments and suggestions, which have greatly helped improve our manuscript. We add one paragraph on page 11 line 343-355 to highlight our key finding. 

“In summary, we have highlighted several key differences between international guidelines and current clinical practice in Taiwan. First, only 30% of units provide protein-containing solutions as initial fluid. This is a systemic issue that requires more resources to improve. Secondly, over half of the neonatal units provided calcium, phosphate, and magnesium at less than the recommended dosage. The inadequate mineral intake in PN may contribute to premature osteopenia and increased fracture risk. Promoting the use of organic calcium and phosphate salts in PN and revising the existing guideline in Taiwan may improve the compliance. Finally, the practice of slow advancement of feeding is common in Taiwan, possibly due to the concerns of the risk of NEC. This feeding policy may lead to prolonged PN usage and extended hospital stays [45,47]. In our opinion, rather than applying a uniform slow feeding approach for all infants, feeding advancement should be customized based on clinical markers, such as gut function and tolerance. ”

Comment 3: There could also be some suggestions from the authors in the discussion how the variation in practice and some of the practices that lead to non compliance could be addressed nationally within Taiwan.

Response 3: Thanks for the great suggestion to improve our manuscript. We do provide some suggestion to improve the compliance in the newly added paragragh as response 2 (page 11 line 343-355) as above.

Reviewer 2 Report

Comments and Suggestions for Authors

Thank you for the opportunity to review the article entitled “Adherence to Nutritional Practice Guidelines in Preterm Infants: A Nationwide Survey in Taiwan.”

The details related to the validation of the questionnaire, complications encountered in premature babies based on the type of feeding, and text corrections need to be specified:

- Validation of the questionnaire: It is not mentioned whether the questionnaire used for data collection was validated or if it was adapted from other sources.  I recommend including details regarding this process or, if validation was not conducted, noting this limitation in the methodology section.

- Correction: In Table 1, in the first row, the term "Level II" should be corrected.

- The authors mention that human milk was the preferred option for EN in preterm infants, followed by donor milk or formulas if human milk was unavailable.  However, the exact types of milk formulas used for preterm infants are not clearly specified.

- Complications related to PN or EN: I suggest clarifying the specific complications associated with PN / EN encountered in the preterm infants included in the study.

Comments on the Quality of English Language

 Minor editing of English language required.

Author Response

Comment 1: Validation of the questionnaire: It is not mentioned whether the questionnaire used for data collection was validated or if it was adapted from other sources.  I recommend including details regarding this process or, if validation was not conducted, noting this limitation in the methodology section.

Response 1 : Thank you for the insightful feedback. We appreciate the time and effort you put into reviewing our work. This questionnaire aid to evaluate the variation of nutrition policy in neonatal units and designed in accordance with the ESPGHAN guidelines and clinical practices, as described in material and method. (page 2, line 58) Since the survey is focus on the clinical practice, therefore there is no validation process for this questionnaire. However, we do agree that the questionnaire may have some limitation, as described in page 11 line 365-371:

“There have been significant advancements in the clinical care of nutrition for preterm infants recently, but some practices have not yet reached a consensus or remain controversial. This questionnaire was developed in accordance with the ESPGHAN guidelines and focuses solely on the general principles of PN and EN care. Some issues of care that have not yet reached a consensus or remain controversial are not included in this survey. Some topics that have not yet reached a consensus in guideline were not included in this survey, such as the use of probiotics or in NICU.  ”

Comment 2: Correction: In Table 1, in the first row, the term "Level II" should be corrected.

Response 2: Thank you for pointing this out. We have corrected the table 1.

Comment 3: The authors mention that human milk was the preferred option for EN in preterm infants, followed by donor milk or formulas if human milk was unavailable.  However, the exact types of milk formulas used for preterm infants are not clearly specified.

Response 3: Thank you for the thorough review. Your observations have been very helpful in enhancing the quality of our paper. The formulas used for preterm infants are 24 kcal/fl oz premature formula if human milk was unavailable (manuscript was modified on page 10, line 292-294). In addition, some units will use 20 cal/fl oz premature formula as initial feeding, which we described the result in page 8, line 177-178:

“Additionally, half of the level II units chose 20 kcal/fl oz premature formula as their second-line option.”

Comment 4: Complications related to PN or EN: I suggest clarifying the specific complications associated with PN / EN encountered in the preterm infants included in the study.

Response 4: Thank you for your thoughtful comments. In this survey, we focus on the clinical practice about the use of PN and EN use in preterm infants and evaluated the nutrition protocol of included units. Therefore, the associated outcome including the complications related to PN or EN were not included in this study. We have surveyed the monitoring schedule in preterm neonates using PN as represented in table 3, and the associated complications was explained on page 6, line 151-154 as below:

“ Some complications may arise during the use of parenteral nutrition, such as hypoglycemia or hyperglycemia, electrolyte imbalance, hyperlipidemia, and PN associated liver disease. Therefore, regular monitoring of associated parameters is suggested.” 

Metabolic bone disease of prematurity is another common preterm comorbidity associated with nutrition intake. The monitor policy for metabolic bone disease of prematurity is presented in table 5.

Reviewer 3 Report

Comments and Suggestions for Authors

The article is overall clear and well-presented. It describes neonatal nutritional practices for preterm infants, particularly very low birth weight (VLBW) infants, in Taiwan, aiming to compare and promote consensus with standardized protocols. The article is appropriately written, and the language is easy to understand. The comparisons are made with recent international guidelines, and the references are accurate. The tables are well-presented, although some sections might be difficult to interpret. Additionally, in Table 1, the second column should refer to "Level II Units" (not Level III).   The conclusions are generally consistent with the arguments presented and are supported by the results. However, the final part of the "Discussion" highlights some limitations of the study, such as the fact that adherence to protocols can sometimes be "operator-dependent," even within the same Neonatal Care Unit, which can skew some results. It would be better to standardize the data as much as possible. Overall, the article is interesting because it emphasizes the need for greater attention to providing the right levels of minerals and nutrients to preterm infants and accelerating the advancement of enteral feeding to prevent certain pathologies, particularly metabolic bone disease of prematurity (MBDP).

Comments on the Quality of English Language

Some ortographic mistakes, such as in the last lines of the "Introduction": "policyess" instead of policies.  

Author Response

Comment 1: Additionally, in Table 1, the second column should refer to "Level II Units" (not Level III).

Response 1: Thank you for pointing this out. We have corrected the table 1.

Comment 2: However, the final part of the "Discussion" highlights some limitations of the study, such as the fact that adherence to protocols can sometimes be "operator-dependent," even within the same Neonatal Care Unit, which can skew some results. It would be better to standardize the data as much as possible.

Response 2: We sincerely appreciate your valuable comments and suggestions, which have greatly helped improve our manuscript. “ However, the response from each unit represents the nutritional policy from the consensus formed among their medical staff. The result reflects the clinical practice of the majority of healthcare professionals. If the survey is collected on a per-person basis, the clinical practice of neonatal units with fewer doctors may be overlooked. ” The above consideration was explained in the discussion on page 11, line 359-362

Comment 3: Some orthographic mistakes, such as in the last lines of the "Introduction": "policyess" instead of policies. 

Response 3: Thank you for the thorough review. We have corrected the spelling errors.